# Well-defined double hysteresis loop in NaNbO$_3$ antiferroelectrics

Nengneng Luo [1] ✉, Li Ma[1,2], Gengguang Luo[1], Chao Xu[3], Lixiang Rao[4] ✉, Zhengu Chen[1,2], Zhenyong Cen[1], Qin Feng[1], Xiyong Chen[1], Fujita Toyohisa [1], Ye Zhu [3], Jiawang Hong [4], Jing-Feng Li [5] ✉ & Shujun Zhang [6] ✉

Antiferroelectrics (AFEs) are promising candidates in energy-storage capacitors, electrocaloric solid-cooling, and displacement transducers. As an actively studied lead-free antiferroelectric (AFE) material, NaNbO$_3$ has long suffered from its ferroelectric (FE)-like polarization-electric field (*P-E*) hysteresis loops with high remnant polarization and large hysteresis. Guided by theoretical calculations, a new strategy of reducing the oxygen octahedral tilting angle is proposed to stabilize the AFE P phase (Space group Pbma) of NaNbO$_3$. To validate this, we judiciously introduced CaHfO$_3$ with a low Goldschmidt tolerance factor and AgNbO$_3$ with a low electronegativity difference into NaNbO$_3$, the decreased cation displacements and [BO$_6$] octahedral tilting angles were confirmed by Synchrotron X-ray powder diffraction and aberration-corrected scanning transmission electron microscopy. Of particular importance is that the 0.75NaNbO$_3$−0.20AgNbO$_3$−0.05CaHfO$_3$ ceramic exhibits highly reversible phase transition between the AFE and FE states, showing well-defined double *P-E* loops and sprout-shaped strain-electric field curves with reduced hysteresis, low remnant polarization, high AFE-FE phase transition field, and zero negative strain. Our work provides a new strategy for designing NaNbO$_3$-based AFE material with well-defined double *P-E* loops, which can also be extended to discover a variety of new lead-free AFEs.

Antiferroelectrics (AFEs), in which the spontaneous polarization dipoles are aligned in opposite directions in adjacent sublattices, can switch to parallel direction under electric field, and return to their original after field removal, leading to double polarization–electric field (*P–E*) hysteresis loop and sprout-shaped strain-electric field (*S−E*) curve[1,2]. The unique characteristics make them promising candidate in a wide range of applications including high-energy-storage capacitors[3–6], electrocaloric solid-cooling[7,8] and high-strain actuators

or transducers[9]. The prerequisite of the unique macroscopic physical characteristics is a reversible phase transformation between antiferroelectric (AFE) and ferroelectric (FE) phases induced by external stimuli such as electric field, mechanical force, and temperature. The most researched antiferroelectric PbZrO$_3$ (PZ) was reported by Sawaguchi et al. in 1951[1]. Since then, hundreds of AFEs have been discovered over the past 70 years, among which PZ remains the prototype for studying the antiferroelectric underlying structure while PZ-based AFE

[1]Center on Nanoenergy Research, State Key Laboratory of Featured Metal Materials and Life-cycle Safety for Composite Structures; School of Resources, Environment and Materials; Guangxi University, 530004 Nanning, China. [2]School of Chemistry & Chemical Engineering, Guangxi University, 530004 Nanning, China. [3]Department of Applied Physics, Research Institute for Smart Energy, The Hong Kong Polytechnic University, Hung Hom, Kowloon, Hong Kong, China. [4]School of Aerospace Engineering, Beijing Institute of Technology, 100081 Beijing, China. [5]State Key Laboratory of New Ceramics and Fine Processing, School of Materials Science and Engineering, Tsinghua University, 100084 Beijing, China. [6]Institute for Superconducting and Electronic Materials, Australian Institute of Innovative Materials, University of Wollongong, Wollongong, NSW 2500, Australia. ✉e-mail: luonn1234@163.com; l.x.rao@bit.edu.cn; Jingfeng@mail.tsinghua.edu.cn; shujun@uow.edu.au

material exhibits the most attracting physical properties. Lead-free AFE materials are promising alternatives to PZ-based system from the viewpoint of enriching AFE types, improving physical properties, and addressing environmental concerns. Only a limited number of lead-free AFEs have been reported so far, among which the $AgNbO_3$ (AN) and $NaNbO_3$ (NN) are considered the most important representatives, where $AgNbO_3$ has made exciting breakthroughs showing reversible AFE−FE phase transition and improved energy-storage density[5,10,11]. Typical double hysteresis loop featuring antiferroelectricity, however, has been barely observed in NN, where a classical FE-like $P–E$ loop was often reported, even though it has been suggested to be an AFE based on crystal structure. Great efforts have been made to achieve a well-defined double $P–E$ loop in NN, through a series of approaches including chemical composition modification[12–14], grain size tailoring[15,16], measuring temperature variation[17], and high-temperature electric treatment[18]. However, only pinched $P–E$ loops with high remnant polarization and large hysteresis can be observed at room temperature, making them far from practical application as AFEs.

Detailed structural investigations found that the AFE nature in NN arises from its P phase (Space group Pbma) which coexists with the FE Q phase (Space group P2₁ma)[19–21] or Rhombohedral phase (Space group R3c)[22]. Our Synchrotron X-ray diffraction (SXRD) confirmed the phase coexistence of Q phase with a majority of P phase (69.6%) in the virgin NN, as shown in Fig. 1a. Due to the comparable energy landscapes between AFE P phase and FE Q phase[17], the electric field-induced Q phase will not back-switch after electric field removal, thus leading to a FE-like $P–E$ loop. This suggests that eliminating the FE Q phase and stabilizing P phase are of particular importance to obtain a well-defined double $P–E$ loop in NN. The structural models of P and Q phases along the b ([010]ₚ) axis are respectively illustrated in Fig. 1b-c. In the P phase, the octahedral tilt can be written unambiguously as a⁻b⁺a⁻/a⁻b⁻a⁻, which is repeated every four octahedras[23]. While, it exhibits a simpler octahedral tilting sequence of a⁻b⁺a⁻ in the Q phase. The images in Fig. 1d-f using bright field (BF) scanning transmission electron microscopy (STEM) further confirms the octahedral rotation behavior[24]. The distinct octahedron rotations result in a quadrupling and doubling of the unit cell size along the b axis for P and Q polymorphs, respectively, as identified in the superlattice reflections at n/4 for P and n/2 for Q in Supplementary Fig. 1. The rotations of P and Q polymorphs are associated with the B-site cation shift, which can be revealed by the displacement derivatives derived from geometric phase analysis (GPA) of the corresponding atomic resolution STEM images[25]. The P phase exhibits characteristic AFE order with anti-parallel atomic shift along [100]p with a periodicity of $4a_p$ (Fig. 1g), while the Q phase exhibits uniform displacement spanning over $10a_p$ indicating the long-range FE order (Fig. 1h).

Previous experimental and theoretical investigations demonstrated the octahedral rotation with the associated cation displacements plays a critical role in stabilizing an AFE structure[26,27]. X-ray/neutron-diffraction on NN showed that the volume fraction of P phase increases with increasing temperature, accompanied by a decreasing amount of Q phase[20]. It is of particular importance that both average cations off-center displacements and oxygen octahedral tilting angles decrease in P phase with increasing temperature, which shows a contrary trend in the Q phase (Supplementary Fig. 2)[20]. It should be noted that in addition to the temperature associated phase transition, the same phenomenon (decrease of octahedral tilting angle) was also observed in modified AN with improved AFE stability[5,28]. To understand the structural distortion for the stability of P and Q phases in NN, density functional theory (DFT) calculations were performed based on method of intermediate configurations from ref. [29]. Figure 1i presents how the free energy (△E is the free energy difference between P and Q

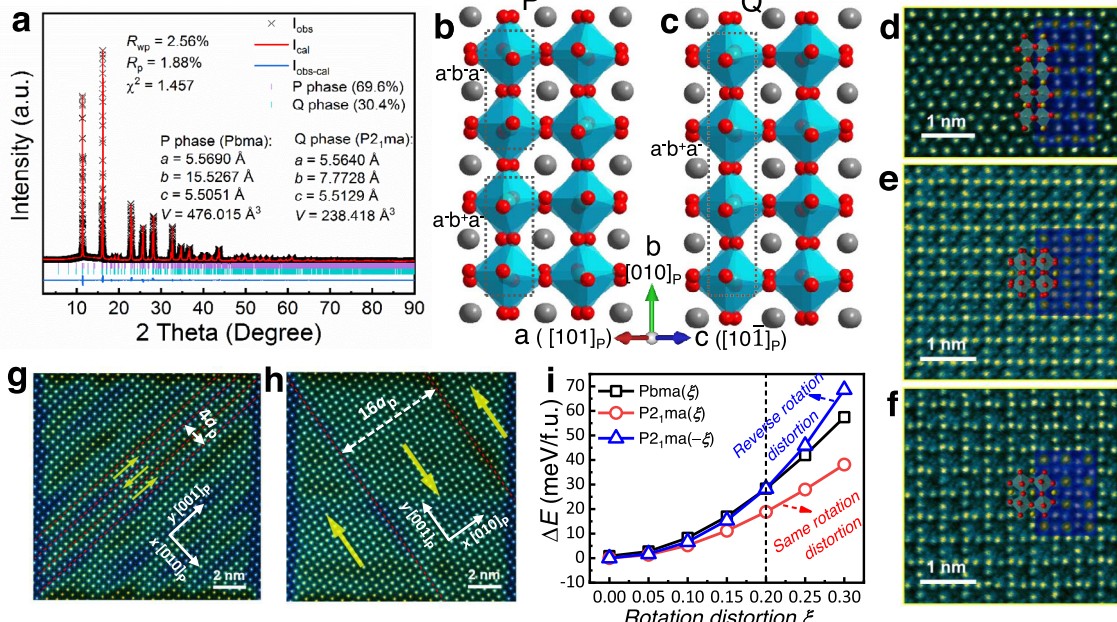

**Fig. 1 | Structure and DFT calculations of NN. a** Rietveld-refinement plot and the refined results of SXRD. Schematic crystal structures of the (**b**) P phase and **c** Q phase. **d** BF STEM image taken from [101]ₚ zone axis, showing the same tilt mode of P and Q phases. **e** BF STEM image taken from [010]ₚ zone axis showing P phase structure. **f** BF STEM image taken from [010]ₚ zone axis showing Q phase structure. The annular dark field (ADF) images and structural schematics are overlaid in (**d**), (**e**), and (**f**), respectively. **g** ADF STEM image taken from [100]ₚ zone axis showing P phase structure. **h** ADF STEM image taken from [100]ₚ zone axis showing Q phase structure. The corresponding GPA shear strain ($\frac{\partial u_y}{\partial x}$) of the ADF STEM images is also overlaid in (**g**) and (**h**), respectively. The sign change of shear strain indicates the reversal of polarization. **i** DFT calculations of free energy as a function of ξ. Based on the continuous parameter ξ, the linear interpolation of atomic positions and lattice parameters was adopted to construct a series of intermediate structures, from P or Q phase (ξ = 0) to the corresponding high symmetry structure (ξ = 1, space group Pm-3m, No. 221). The parameter ξ reflects the rotation distortion of the structure, i.e., there is no distortion at ξ = 1, while distortion becomes stronger with smaller ξ and even stronger with ξ < 0.

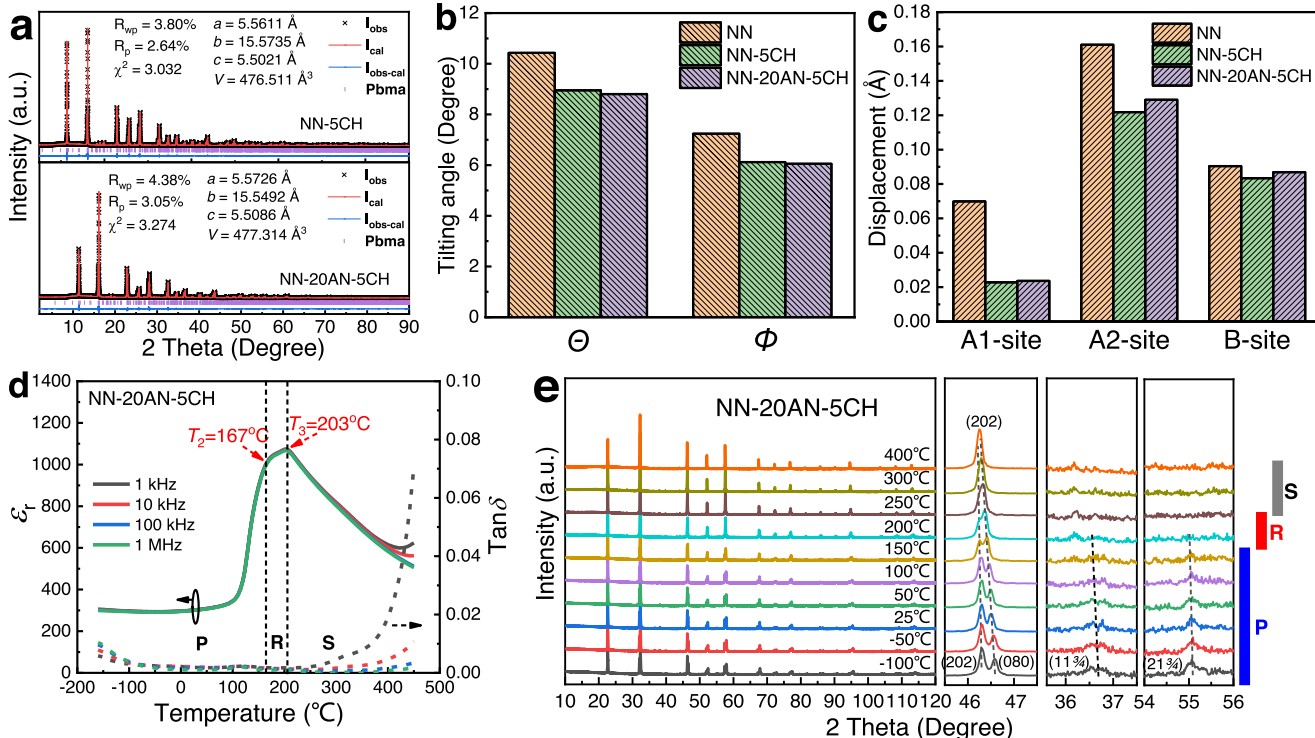

**Fig. 2 | SXRD, dielectric properties and temperature dependent XRD.**
**a** Retrieved refinement plots of SXRD for NN-5CH and NN-20AN-5CH. A comparison study of (**b**) the [BO₆] octahedral tilting angles $\Theta$ and $\Phi$, and (**c**) the displacement of A1, A2, and B-site cations for NN, NN-5CH and NN-20AN-5CH ceramics derived from the Rietveld refinement of SXRD. Temperature dependence of (**d**) dielectric permittivity and loss, and (**e**) XRD for NN-20AN-5CH ceramic.

phases) varies as the lattice structure deviates from P(Q) phase. The P phase has slightly higher total energy than Q phase ($\triangle E = 0.86$ meV/f.u.), being consistent with the previous report[17]. As the rotation distortion of P phase becomes smaller (larger $\xi$), the energy curve of P phase intersects with Q phase whose rotation distortion becomes larger (smaller $-\xi$). This suggests that the P phase can be more stable than Q phase if the former becomes less distorted while the latter more distorted, for example, the scenario at elevated temperature, as reported in previous work[20]. It shows that the reverse rotation distortion change tunes the stability competition of P and Q phases and leads to Q-to-P phase transition, while the same rotation distortion change does not, as the red Q phase curve is always below P phase curve with the same distortion change, as shown in Fig. 1i. This also suggests that reducing the rotation distortion may be beneficial to stabilize the P phase, while increasing the rotation distortion of the co-existing Q phase by other approaches, for example, doping method in this work, will also stabilize the P phase, that said, facilitating the Q-to-P phase transition.

To experimentally verify the feasibility and effectiveness of our proposed idea, a new material system with composition of (1-x-y)NaNbO₃-xAgNbO₃-yCaHfO₃:1.5 mol% MnO₂ (NN-100xAN-100yCH) was designed and prepared. According to previous experiments on NN, the addition of a component with smaller Goldschmidt tolerance factor (t) can stabilize the AFE P phase[12–14,17]. Therefore, the addition of CaHfO₃ with a lower t (CH: 0.9182; NN: 0.9671) is considered to be an effective approach to reduce the oxygen octahedral tilt[30]. The AN is also used to stabilize AFE phase in NN due to its lower electronegativity difference (1.675, versus 2.175 for NN). In addition, the MnO₂ is used as a sintering aid to improve the sintering ability and reduce the defects and leakage current[31]. Herein, we mainly focus on NN, NN-20AN, NN-5CH, NN-5CH-20AN to investigate the structure evolution and the corresponding AFE properties. SXRD and STEM revealed that the addition of AN and/or CH can significantly reduce the oxygen octahedral tilting angles, as well as cation displacements. Of particular importance is that typical

AFE features, well-defined double P–E loops and sprout-shaped S–E curves, were successfully achieved in NN-5CH-20AN.

## Results

### Structure of NN-100xAN-100yCH ceramics

To understand the structure evolution after AN/CH modification, we investigated the XRD of the NN-100xAN-100yCH ceramics. It is found that the addition of a certain amount of AN or CH can make NN maintain an orthorhombic AFE P structure, while a large amount of CH (>7 mol%) will lead to an orthorhombic AFE R phase (Space group Pnma) (Supplementary Fig. 3)[14,17]. The Rietveld refinements of high-energy SXRD for NN-5CH and NN-20AN-5CH are well-fitted to the P phase without detectable Q phase (Fig. 2a), indicating that the CH or AN/CH can effectively eliminate Q phase. On the basis of Rietveld-refinement results in Supplementary Table 1, we make a comparative study of the [BO₆] octahedral tilt and the cation off-centering displacements among NN, NN-5CH and NN-20AN-5CH, as shown in Fig. 2b, c, respectively. The [BO₆] octahedral tilting angles $\Theta$ about $a$ axis and $\Phi$ about $b$ axis (Schematics of $\Theta$ and $\Phi$ are presented in Supplementary Fig. 4) are 10.13° and 7.24° for NN, which significantly decrease to 8.95° and 6.12° for NN-5CH, and the addition of AN further enhances this trend. This is believed to be a result of the synergistic effect of the lower tolerance factor of CH and the stronger covalent bond characteristic of Ag–O bond in AN. Due to a coupling effect, the change of octahedral rotation also leads to the change of cation off-centering displacements. The NN-5CH and NN-20AN-5CH exhibits obviously reduced displacements of A1 and A2-site cations compared to those of NN (Schematics of A1, A2 and B-site cations are also shown in Supplementary Fig. 4). Even though the displacement change of B-site ions is not as obvious as that of A-site ions, a decrease still can be observed for the modified compositions. The addition of AN in NN-5CH slightly increases cation displacements, which may be associated with the higher polarizability of Ag⁺ compared to Na⁺. The reduced octahedral tilting angles and cation displacements are believed to benefit the stabilization of AFE structure.

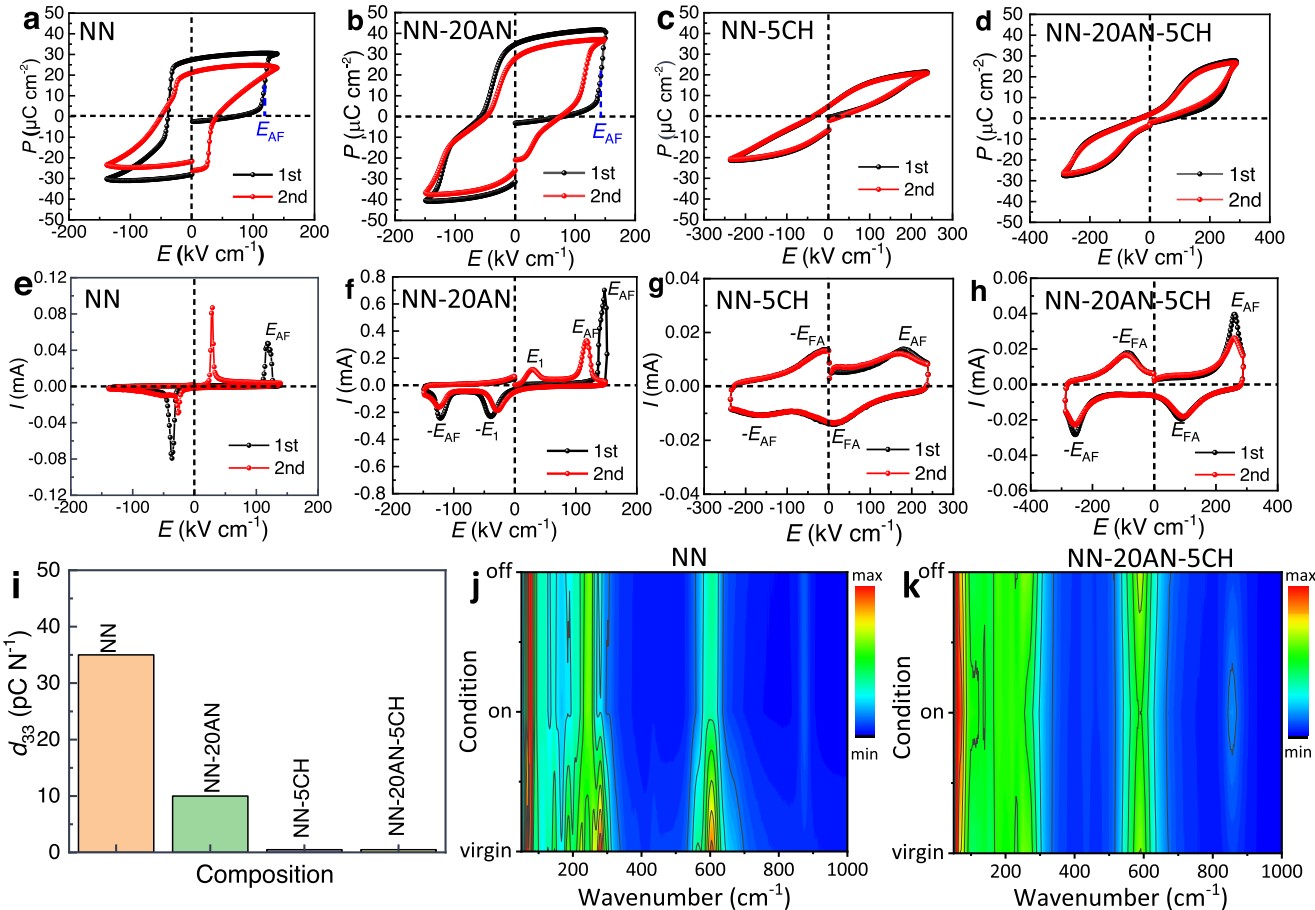

**Fig. 3 | The reversible behavior of the newly designed NN-based ceramics.** The first and second circles of P−E loops and I−E curves for (**a**) and (**b**) NN; (**c**) and (**d**) NN-20AN; (**e**) and (**f**) NN-5CH; (**g**, **h**) NN-20AN-5CH. **i** The piezoelectric coefficient $d_{33}$ for the poled samples. The Raman spectra under various conditions of virgin state, electric field on and off states for (**j**) NN and (**k**) NN-20AN-5CH. The electric fields applied are 150 and 300 kV cm$^{-1}$ for NN and NN-20AN-5CH respectively, to ensure the field-induced AFE−FE phase transition.

The additions of CH and AN also change the temperature-induced phase transition behavior in NN, as depicted in the temperature-dependent dielectric spectra of NN-100xAN-100yCH ceramics (Supplementary Fig. 5, 6), revealing an altered physical nature. In particular, two dielectric anomalies are observed in NN-20AN-5CH over measuring temperature range (Fig. 2d), with a shoulder peak locating around 167 °C (marked as $T_2$) and the other one locating around 203 °C (marked as $T_3$). To reveal the phase structure evolution sequence of NN-20AN-5CH, temperature-dependent XRD was measured and given in Fig. 2e. The XRD is found to be in good agreement with AFE P phase over a large temperature range from -100 to -170 °C, AFE R phase around 170–250 °C, and paraelectric (PE) S phase (Space group Pbnm) over 250 °C, based on the Rietveld refinements (Supplementary Fig.7). The temperature dependences of the lattice parameters and cell volume further confirm the phase transition. Therefore, it can be deduced the $T_2$ is associated with AFE P to AFE R phase transition, while $T_3$ is assigned to be AFE R to PE S phase transition temperature, even though with small variation in temperature with XRD analysis.

## Reversibility between AFE and FE phases

Reversibility of the electric field-induced AFE−FE phase transition is essential for an AFE material[9]. To evaluate the reversible behavior of the newly designed compositions, we measured the first and second cycles of the P−E loops and I−E curves by using NN, NN-20AN, NN-5CH, and NN-20AN-5CH as representatives, as shown in Fig. 3. A slowly increasing polarization at low electric field is observed in virgin NN ceramic, exhibiting a sudden improvement when the electric field increases to a critical value (Fig. 3a) accompanied by a current peak (marked as $E_{AF}$) in the I−E curve (Fig. 3b). The sudden change in polarization and current is associated with the AFE to FE phase transition. However, a large remnant polarization remains when the electric field reduces to zero, due to the irreversible FE to AFE phase transition. The typical square shaped P−E loop and two-peak I−E curve in the second cycle confirm a FE nature for the high electric-field-treated sample, in accordance with those generally observed in NN ceramic reported earlier[32]. After partially substituting Na$^+$ with Ag$^+$, the first positive half cycle of P−E loop is similar to that of virgin NN, while the first negative half cycle and second cycle of the loops exhibit pinched shape, as shown in Fig. 3c. Meanwhile, the first positive half cycle of I−E curve exhibits one current peak similar to that of virgin NN, while two peaks are observed in the negative half part (Fig. 3d). It should be noted that four peaks are observed in the second cycle of I−E curve. However, in contrast to the classical I−E curve in an AFE material, where two peaks appear at increasing electric field and the other two upon withdrawal, all these four peaks appear upon increasing electric field. The peaks appearing at higher electric field are associated with the AFE−FE phase transition, analogous to that usually observed in an AFE material. The other two peaks appearing at lower electric field, however, may be caused by the induced FE phase back-switching to AFE phase (to differ them from the so-called reversible FE−AFE phase transition, we define this critical electric field as $E_1$ and $-E_1$). Due to the irreversibility of FE−AFE phase transition in NN-20AN, the induced FE phase cannot spontaneously reverse to the AFE state, thus a backward electric field is required to overcome the energy barrier. Of particular

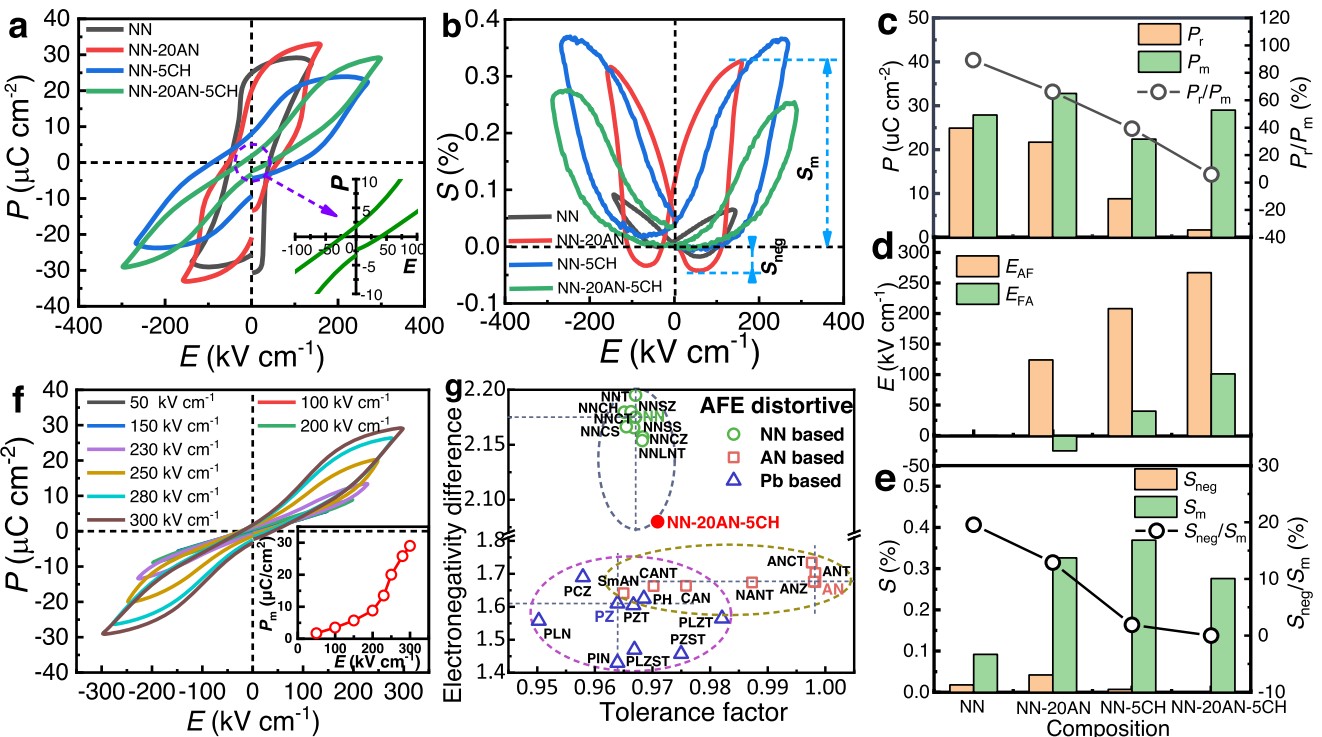

**Fig. 4 | The polarization and strain of the newly designed NN-based ceramics.**
**a** $P–E$ loops and (**b**) $S–E$ curves. The inset of a shows magnified scale for NN-20AN-5CH. Compositions dependence of (**c**) $P_m$, $P_r$, $P_r/P_m$; (**d**) $E_{AF}$, $E_{FA}$; (**e**) $S_m$, $S_{neg}$, $S_{neg}/S_m$, derived from the $P–E$ loops, the $I–E$ curves in Supplementary Figs. 12 and 13, and the $S–E$ curves, respectively. **f** Electric field dependence of $P–E$ loops for NN-20AN-5CH, the inset gives the $P_m$ as a function of maximum measuring electric fields. **g** The electronegativity difference and tolerance factor of NN-based[12,13,17,46–50], AN-based[5,51–56] and Pb-based[8,26,57–62], AFEs. (NNT: NaNb$_{0.6}$Ta$_{0.4}$O$_3$; NNSZ: Na$_{0.96}$Sr$_{0.04}$Nb$_{0.96}$Zr$_{0.04}$O$_3$; NNCH: Na$_{0.96}$Ca$_{0.04}$Nb$_{0.96}$Hf$_{0.04}$O$_3$; NNCT:

Na$_{0.9}$Ca$_{0.1}$Nb$_{0.9}$Ti$_{0.1}$O$_3$; NNCS: Na$_{0.96}$Ca$_{0.04}$Nb$_{0.96}$Sn$_{0.04}$O$_3$; NNSS: Na$_{0.95}$Sr$_{0.05}$Nb$_{0.95}$Hf$_{0.05}$O$_3$; NNCZ: Na$_{0.96}$Ca$_{0.04}$Nb$_{0.96}$Zr$_{0.04}$O$_3$; NNLNT: Na$_{0.95}$La$_{0.05}$Nb$_{0.9}$Ti$_{0.1}$O$_3$; ANT: AgNb$_{0.45}$Ta$_{0.55}$O$_3$; CANT: Ag$_{0.9}$Ca$_{0.05}$Nb$_{0.95}$Ta$_{0.05}$O$_3$; NANT: Ag$_{0.97}$Nd$_{0.01}$Ta$_{0.20}$Nb$_{0.80}$O$_3$; ANZ: AgNb$_{0.99}$Zr$_{0.01}$O$_{2.995}$; ANCT: Ag$_{0.98}$Ca$_{0.02}$Nb$_{0.98}$Ti$_{0.02}$O$_3$; SmAN: Ag$_{0.91}$Sm$_{0.03}$NbO$_3$; CAN: Ag$_{0.92}$Ca$_{0.04}$NbO$_3$; PH: PbHfO$_3$; PZT: PbZr$_{0.95}$Ti$_{0.05}$O$_3$; PCZ: Pb$_{0.88}$Ca$_{0.12}$ZrO$_3$; PIN: PbIn$_{0.5}$Nb$_{0.5}$O$_3$; PLN: PbLu$_{0.5}$Nb$_{0.5}$O$_3$; PLZT: (Pb$_{0.97}$La$_{0.02}$)(Zr$_{0.55}$Ti$_{0.45}$)$_{0.995}$O$_3$; PLZST: Pb$_{0.97}$La$_{0.02}$Zr$_{0.5}$Sn$_{0.45}$Ti$_{0.05}$O$_3$; PZST: PbZr$_{0.455}$Sn$_{0.455}$Ti$_{0.09}$O$_3$.

importance is that the double $P–E$ loops and four-peak $I–E$ curves are observed for both NN-5CH and NN-20AN-5CH, which almost overlap on first and second cycles, as shown in Fig. 3e-h, demonstrating typical AFE characteristics and good reversibility of AFE−FE phase transitions. The reversibility of the induced FE phase can also be reflected by the piezoelectric coefficient ($d_{33}$), with values around 35 and 10 pC N$^{-1}$ in poled NN and NN-20AN respectively (Fig. 3i), clearly demonstrate the irreversible field-induced FE phase in these compositions[33]. In contrast, nearly zero $d_{33}$ is observed for both NN-5CH and NN-20AN-5CH, revealing the field-induced FE phase switches back to AFE state upon field removal[9,34]. Interestingly, the NN-20AN-5CH exhibits the least sensitive $\varepsilon_r$ under electric field, where the $\varepsilon_r$-$E$ curves almost overlap before and after field application (Supplementary Fig. 8), indicating the highest stability of its AFE phase.

The good reversibility between the AFE and field-induced FE phases in NN-20AN-5CH can also be corroborated by the in-situ electric field-dependent Raman spectra. At a low electric field, both NN and NN-20AN-5CH maintain characteristics of the AFE P phase (Supplementary Fig. 9). A high electric field above 90 kV cm$^{-1}$ gradually induces AFE P to FE Q phase transition in NN, while a significantly improved electric field above 200 kV cm$^{-1}$ is required to induce P-to-Q phase transition in NN-20AN-5CH. It is interesting that the Raman spectra of NN remain almost the same after electric field removal, as shown in Fig. 3j. This reflects that the electric field-induced FE Q phase maintains and cannot switch backward to AFE P phase upon the removal of electric field, showing an irreversible Q-P phase transition behavior. On the contrary, the Raman spectra of NN-20AN-5CH return to almost the same shape as the original state after electric field removal (Fig. 3k), demonstrating good reversibility between P and Q phases.

## $P–E$ loops, $S–E$ curves and the possible mechanism

Figure 4a-b display the $P–E$ loops and $S–E$ curves of the four representative compositions, respectively. The pure NN exhibits a square-shaped $P–E$ loop with high remnant polarization and large hysteresis, as well as a butterfly-shaped $S–E$ curve, showing the characteristics of a ferroelectric material. The addition of sole AN or CH in NN can transform the $P–E$ loop into a pinched shape, as observed in NN-20AN and NN-5CH. However, the NN-20AN shows a butterfly-shaped $S–E$ curve, similar to that of pure NN but showing higher strain. In contrast, the NN-5CH shows sprout-shaped $S–E$ curve, in which the strain disappears on the removal of the electric field. The difference in shape may be resulted from the increased FE−AFE reversibility for the CH-modified NN. Of particular importance is that a well-defined double $P–E$ loop with near-zero remnant polarization and significantly reduced hysteresis, accompanying with sprout-shaped $S–E$ curve, can be achieved when adding appropriate CH and AN (such as NN-20AN-5CH), demonstrating typical AFE characteristics.

To quantitatively evaluate the strength of antiferroelectricity, we make a comparative study on the compositional dependence of $P_m$ (the maximum polarization), $P_r$, $P_r/P_m$, $E_{AF}$, $E_{FA}$, $S_m$ (the highest stain), $S_{neg}$ (the negative strain, namely the difference between the zero-field strain and the lowest strain), and $S_{neg}/S_m$ of the four compositions, as shown in Fig. 4c-e. All samples exhibit high $P_m$ with value in the range of 20-30 μC cm$^{-2}$, indicating the formation of long-range ferroelectric phase under high electric field. The $P_r$ decreases significantly from 24.9 μC cm$^{-2}$ for NN to 1.7 μC cm$^{-2}$ for NN-20AN-5CH, leading to an obvious reduction of $P_r/P_m$ from 89 to 7%. The $E_{AF}$ and $E_{FA}$ cannot be detected for NN, both of which increase obviously after AN and CH modification. It is worth noting that NN-20AN-5CH exhibits significantly high $E_{AF}$ of

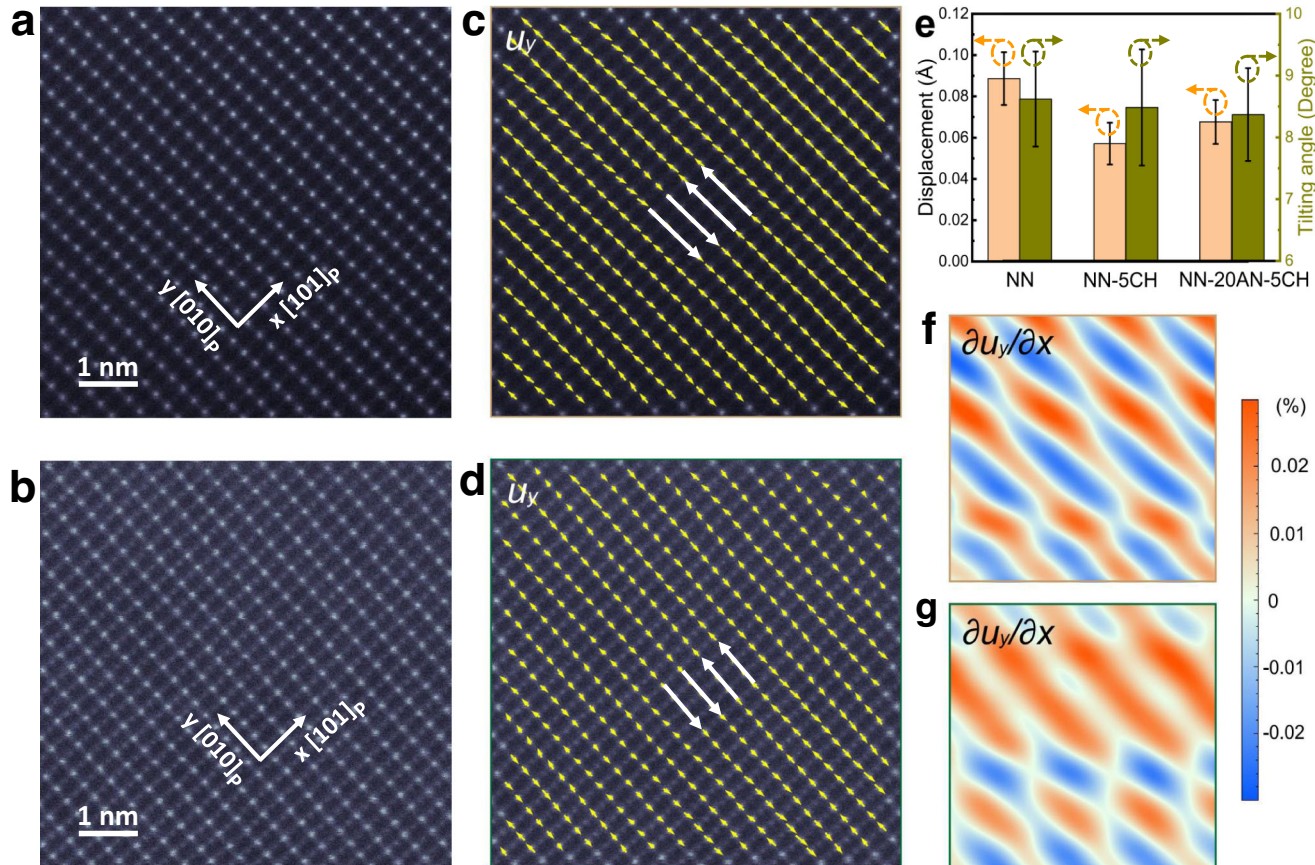

**Fig. 5 | Polarization field mapping.** ADF images for (**a**) NN and (**b**) NN-20AN-5CH. B-site displacement map for (**c**) NN and (**d**) NN-20AN-5CH. The long white lines in (**c**) and (**d**) with antiparallel vectors are guiding line for revealing B-site displacements. **e** Displacement averaged from (**c**) and (**d**), oxygen octahedral tilting angle $\Theta$ averaged from Supplementary Fig. 19 for NN and NN-20AN-5CH. **f** Displacement derivative $\frac{\partial u_y}{\partial x}$ from (**a**) indicating displacement wave of NN using geometric phase analysis (GPA). **g** Displacement derivative $\frac{\partial u_y}{\partial x}$ from (**b**) indicating displacement wave of NN-20AN-5CH using GPA.

267 kV cm$^{-1}$ and $E_{FA}$ of 101 kV cm$^{-1}$, which are comparable with those of well-known PZ- and AN- based AFEs. On the other hand, the $S_m$ increases significantly after the addition of AN or CH, while the $S_{neg}$ goes to zero, leading to an obvious decrease of $S_{neg}/S_m$ for NN-5CH and NN-20AN-5CH. All these merits clearly demonstrate that the NN-20AN-5CH is a good AFE material with highly stabilized AFE phase. Of particular significance is that the $P$–$E$ loop of NN-20AN-5CH exhibits the typical changes consistently observed in AFEs with an increasing electric field (Fig. 4f), good frequency stability in the range from 1 to 300 Hz (Supplementary Fig. 10), as well as good aging-related performance over 180 days (Supplementary Fig. 11).

As discussed above, the addition of AN and CH can significantly improve the stability of the AFE P phase, this concept can be further strengthened by the CH and AN dependence of $P$–$E$ loops, $I$–$E$ and $S$–$E$ curves as shown in Supplementary Figs. 12 and 13, as well as the changes in electrical properties and stain listed in Supplementary Table 2. It should be noted that the strong antiferroelectricity can be ruled out from the effect of micromorphology, because both NN and CH-modified NN-based ceramics exhibit similar gain size and morphology, though a small amount of Hafnium-rich oxide is observed in CH-modified NN-based ceramics (Supplementary Figs. 14 and 15). It is readily accepted that the CH is beneficial for stabilizing AFE phase of NN, due to its lower tolerance factor in comparison with NN. However, this cannot explain the effect of AN, since it has larger $t$ (0.9981)[30]. This means the tolerance factor is not the only reason that accounts for the improved AFE stability. The electronegativity difference is another factor that should be considered when design an AFE material. From the general observations of electronegativity difference and tolerance factor for PZ-, AN-

and NN-based AFEs given in Fig. 4g, it can be concluded that a good AFE material should possess both low electronegativity difference and tolerance factor. Even though the NN-based materials possess low $t$, but the excessively high electronegativity difference (2.175 for NN) is not conducive to stabilizing AFE phase. In comparison, the AN has much lower electronegativity difference of 1.675[35], which can remarkably reduce the total electronegativity difference in the modified NN system, thus leading to improved AFE stability. Based on this design principle, we have also successfully synthesized several NN-based AFEs with other components, such as 0.72NaNbO$_3$-0.20AgNbO$_3$-0.06CaTiO$_3$ (NN-20AN-6CT) and 0.76NaNbO$_3$-0.20AgNbO$_3$-0.04CaZrO$_3$ (NN-20AN-4CZ). As expected, they exhibit typical AFE $P$–$E$ loops, $I$–$E$ curves and $S$–$E$ curves, as shown in Supplementary Fig. 16.

**Microscopic mechanism based on STEM**

To gain insight into the microscopic mechanism, we conducted ADF STEM along [10$\bar{1}$]$_p$ zone axis to map the polarization field of NN, NN-5CH and NN-20AN-5CH. Due to the ultra-low Z-contrast of the A-site ions, we can only map the positions of the B-site ions, as presented in Fig. 5a, b and Supplementary Fig. 17. It is difficult to precisely measure the polarization based on the ADF images by calculating the relative displacement of B-site ions with respect to A-site ones as generally adopted in perovskite ferroelectrics such as PbTiO$_3$[36]. Herein phase lock-in analysis was performed[37–39], which enables the periodic lattice displacement to be precisely mapped. As shown in Fig. 5c, d, we can see the collective atomic displacement primarily along [010]$_p$ direction, which forms a transverse displacive modulation wave with a wavelength of 4a$_p$. The antipolar arrangement of the lattice displacement gives rise to

satellite diffraction at n/4 positions in the electron diffractions and fast Fourier transform (FFT) of the ADF images (Supplementary Fig. 18), well manifesting the antiferroelectricity. More importantly, the NN shows a significantly larger B-site displacement of -0.089 Å than that of NN-5CH (-0.057 Å) and NN-20AN-5CH (-0.068 Å), as plotted in Fig. 5e, in consistence with the SXRD analysis. This unambiguously demonstrates a heavy suppression of the ferroelectric ordering. The comparison of the atomic displacement derivative can also be observed from the magnitude of shear strain in GPA images, as depicted in Fig. 5f-g. The larger displacement in NN contributes to larger shear strain where polarization changes the direction, which can be reflected by the darker color (Fig. 5f) in comparison to that observed in NN-20AN-5CH (Fig. 5g). Along the $[101]_p$ axis, the antiphase oxygen octahedral tilt is mapped (Supplementary Fig. 19), from which the oxygen octahedral tilting angle $\Theta$ can be calculated, as given in Fig. 5e. Despite large measurement errors, a slight decrease in tilting angle can still be revealed, further confirming the SXRD analysis.

In summary, based on the structure analysis and DFT calculations, we suggest that reducing oxygen octahedral tilt may benefit the stabilization of AFE P phase in NN ceramics. To validate this concept, a series of lead-free NN-based AFEs were synthesized, among which the $NaNbO_3$-$20AgNbO_3$-$5CaHfO_3$ ceramic exhibits well-defined double $P–E$ loops and sprout-shaped $S–E$ curves. XRD and STEM analysis reveal a reduction in oxygen octahedral distortion angle and ionic displacement, which are associated with the low tolerance factor of CH and small electronegativity difference of AN, accounting for the observed stabilized AFE P phase. This may help guide future material design for the discovery of a large number of lead-free AFEs.

## Perspective

This work successfully addresses the long-standing challenge of developing new lead-free $NaNbO_3$-based AFE ceramics with reversible AFE–FE phase transition. The double $P–E$ loops and sprout-like $S–E$ curves of the newly designed compounds are of critical importance for a variety of applications including high-energy-storage capacitors and high-strain actuators. We also proposed an innovative strategy for designing new lead-free AFEs by reducing the distortion angle of the oxygen octahedra. The concept presented here not only provides a practical way of developing NN-based AFEs with reversible AFE–FE phase transition, but can also be employed to discover other lead-free AFE systems. It should be pointed out that even though we obtained reversible AFE–FE phase transition in NN, the hysteresis of the $P–E$ loops and $S–E$ curves cannot be completely eliminated. More work should be done to achieve hysteresis-free $P–E$ loops and $S–E$ curves, which is of great importance for practical use of AFEs.

## Methods
### Theoretical calculations
The Vienna ab initio Simulation Package (VASP) was employed to conduct Density functional theory (DFT) calculations[40,41]. The projector-augmented wave (PAW) method was adopted to describe the ion−electron interaction, while the generalized gradient approximation (GGA) with the Perdew-Burke-Ernzerhof (PBE) functional was used to depict the exchange-correlation part of the electron-electron interaction[42]. A plane wave basis set with a kinetic-energy cutoff of 500 eV was implemented to ensure convergence[43], and crystal structures were optimized until energy and force convergence reached a threshold of <10−6 eV and 0.01 eV Å−1, respectively. In addition, sampling of the Brillouin zone was carried out using the Monkhorst-Pack k-point meshes with a reciprocal space resolution of $2\pi \times 0.03$ Å−1 for structural optimization[44,45].

### Fabrication of ceramics
The $(1-x-y)NaNbO_3$-$xAgNbO_3$-$yCaHfO_3$ with 1.5 mol% $MnO_2$ ($x = 0.2$, $y = 0$, 0.01, 0.02, 0.03, 0.04, 0.05, 0.06, 0.07, abbreviated as NN-

20AN-100$y$CH; $y = 0.05$, $x = 0$, 0.1, 0.2, 0.3, 0.4, abbreviated as NN-100$x$AN-5CH) ceramics were synthesized by a high-temperature solid-state method. The reagents used were high-purity $Na_2CO_3$ (99.8 %), $Nb_2O_5$ (99.99 %), $HfO_2$ (99.99 %), $CaCO_3$ (99.8 %), $Ag_2O$ (99.7 %) and $MnO_2$ (98.8 %). Stoichiometric amounts of reagents were weighted, ball milled, and then calcined at 900-980 °C for 6 h in air or in $O_2$ atmosphere depending on their compositions. The calcined powders were ball milled again with $MnO_2$ addition, dried and hand-pressed into pellets with 7 mm in diameter and 1 mm in thickness. Then, the pellets were cold isostatic pressed again at 250 MPa to further improve their density, followed by sintering at 1190–1320 °C for 5 h in air or in $O_2$ atmosphere. Finally, the as-sintered pellets were polished, coated with silver electrode on both sides, and fired at 600 °C for 30 min.

### Characterization of phase structure
Phase purity and crystal structures of the as-prepared samples were characterized by using an X-ray powder diffractometer (XRD, SmartLab-3kW, Rigaku Ltd., Tokyo, Japan) with Cu Kα radiation. The high-quality synchrotron X-ray powder diffraction (SXRD) data was performed at TPS 19 A (Taiwan Photon Source) of the National Synchrotron Radiation Research Center with a calibrated wavelength of 0.77489 Å and the energy of 16 KeV. The Rietveld analysis of the SXRD was performed using GSAS. The microstructure was observed by using a scanning electron microscope (SEM, FE-SEM Sigma 300, ZEISS Corp., German) with energy dispersive spectroscopy (EDS), after polished and thermally etched. Aberration-corrected scanning transmission electron microscope (STEM) was performed on a probe corrected FEI Spectra 300 S/TEM (ThermoFisher Scientific, Eindhoven, Netherlands) equipped with an X-FEG source and operated at an accelerating voltage of 300 kV. A beam current of 50 pA and a semi-angle of convergence of 24.4 mrad was utilized. ADF images were collected with a detector semi-angle range of 72–200 mrad while BF images were collected with a detector outer semi-angle range of 6 mrad. Atom column locations were determined via 2D Gaussian fitting.

### Measurement of dielectric and piezoelectric properties
The temperature dependence of dielectric permittivity and loss was measured using a precision impendence analyzer (E4990A, Keysight, Bayan, America) connecting with a temperature control system (DMS-500, Balab, Wuhan, China) in the temperature ranging from −150 to 450 °C at frequencies of 1, 10, and 100 kHz. The variation of dielectric permittivity at virgin and poled state was measured before and after poling the samples over their corresponding AFE–FE phase transition electric field. The piezoelectric coefficient $d_{33}$ was recorded by using a Piezo-$d_{33}$ meter (ZJ-3A, Chinese Academic Society, Beijing, China), after poling the samples over their corresponding AFE–FE phase transition electric field. The electric field dependence of normalized dielectric permittivity (dielectric tunability) was measured using a ferroelectric tester (TF Analyzer 3000, aixACCT, Aachen, Germany) with a maximum bias field of 100 kV cm−1.

### Measurement of ferroelectric properties and strain
The polarization vs. electric field hysteresis ($P–E$) loops, current vs. electric field ($I–E$) curves and strain vs. electric field ($S–E$) curves were characterized at 1 Hz using a ferroelectric tester.

## Data availability
The data that support the plots within this paper and other findings of this study are either provided in the Article and its Supplementary Information or available from the corresponding author upon request.

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

## Acknowledgements
This work was financially supported by the National Natural Science Foundation of China (Grant No. 52072080) and Guangxi Natural Science Fund for Distinguished Young Scholars (Grant No. 2022GXNSFFA035034). Y.Z. acknowledges the support of the Research Grants Council of Hong Kong (Grant No. C5029-18E). J.L. acknowledges the support of Tsinghua-Foshan Innovation Special Fund (TFISF) under Grant No. 2020THFS0113.

## Author contributions
N.L. conceived the initial concept. N.L., G.L. and L.M. prepared the samples and processed the experimental data. J.H. and L.R. conducted the theoretical calculations. C.X. and Y.Z. conducted the TEM observations. Zhengu Chen, Zhenyong Cen, and Q.F. assisted to measure the electrical properties. X.C. and T.F. helped to analysis the crystal structures. N.L., J.-F.L., and S.Z. interpreted the theoretical and experimental results. N.L. and S.Z. wrote the paper, all authors discussed and edited the paper.

## Competing interests
The authors declare no competing interests.
