## [Peer Review File · Nature Communications]

Well-defined double hysteresis loop in NaNbO₃ antiferroelectricsREVIEWER COMMENTS

Reviewer #1 (Remarks to the Author):

The manuscript by Nengneg Luo et al. demonstrate a well-defined double hysteresis loop in NaNbO_3 antiferroelectrics ceramics by doping with CaHfO_3 and AgNbO_3 to stabilize the AFE P phase. The beautiful STEM images clearly confirmed the decreased cation displacements and BO_6 octahedral tilting angles, validating the theoretical prediction. The results are very interesting and important. I think that the manuscript is publishable in Nature Communications. Some small concerns are as follows:

- (1) The behaviors of the P-E hysteresis loops of the ceramic samples also depend on their grain size and morphology. The doping of AgNbO_3 and CaHfO_3 might change their microstructure obviously. In order to draw a solid conclusion, the authors should exclude the effect of grain morphology on its FE and AFE properties. So it is better to provide some SEM images or low mag TEM images.
- (2) Do the doping elements (Ag, Ca and Hf) distribute uniformly? Is there any nanoscale segregation or phase separation?
- (3) I want to know if the BF-STEM images shown in Fig. 1 d-f have been processed. The image contrast is a little strange for me.
- (4) In the caption of Fig.1, "The ADF images and structural schematics are overlaid in a, e, and f, respectively." should be "are overlaid in d, e, and f, respectively."

Reviewer #2 (Remarks to the Author):

The present work deals with the "design" (tailoring) of NaNbO_3 based ceramics with the aim of enabling field induced reversible phase transformations between the antiferroelectric and ferroelectric phases (which manifests itself in a corresponding well defined P-E hysteresis curve).

NaNbO_3 shows promising properties among lead-free antiferroelectric materials. However, although it has a seemingly simple chemical composition, it has long been regarded as probably the "most complex perovskite system known". This is due to the complex NbO_6 octahedral tilts that occur in the unit cell. These lead to a large number of different, partly (anti-)ferroelectric phases, with corresponding phase transitions between them. A special feature of NaNbO_3 is that at room temperature the energy difference between the ferroelectric Q-phase and the antiferroelectric P-phase is so small that both phases coexist and field-induced phase transformations can easily occur. In the case of NaNbO_3 , however, the application of an electric field leads to field-induced irreversible formation of the ferroelectric Q phase, resulting in a (broad) P-E hysteresis curve with a very high remanence. This property has so far prevented the technological use of pure NaNbO_3 as an antiferroelectric material.

This was the starting point of the present work. With the deliberate addition of AN (AgNbO_3) and CH (CaHfO_3), the NbO_6 tilts were reduced and thus the antiferroelectric P-phase could be stabilized. Moreover, the authors were able to demonstrate the reversible behavior of the such tailored ceramics in the P-E and I-E hysteresis curves. The (macroscopic) electrical investigations are thereby complemented by a very extensive structural characterization by means of X-ray diffraction and electron microscopy and supported by corresponding DFT calculations.

In my opinion, the authors impressively demonstrate with the present work the potential of the nowadays existing possibilities to tailor a material. This involves both pre-consideration of how the material should be designed to achieve the desired structural and physical properties, and the application of a variety of experimental techniques to verify the actual properties. Therefore, I believe that the present manuscript can be published in Nature Communications if the authors implement the following revisions.

(i) In the literature, the coexistence of long-range antiferroelectric and ferroelectric phases is – on a microscopic scale - discussed to be caused by irregularly arranged nanotwins resulting from complex local octahedral tilt patterns (e.g., Johnston 2010, <https://doi.org/10.1021/ja101860r> ; Jiang 2013, <https://doi.org/10.1103/PhysRevB.88.014105>). Can you please comment on this for the present material system?

(ii) The manuscript is written very compactly, i.e. the reader has to "process" very much information in a few sentences. Due to this high data density (especially also in the supplementary), one has to search very often in the text for the corresponding figures. In my opinion, this makes the flow of the text somewhat difficult. The high density of information also means that the figures often appear overloaded and individual diagrams are sometimes also too small to still be able to recognize important details. I would therefore recommend to revise the illustrations in this respect and also to think about a corresponding revision of the text. In addition, the manuscript contains occasional language errors, but these can be easily corrected.

Minor comments

Besides occasional linguistic errors I found the following minor errors:

- i. Line 145: “.. are overlaid in d, e, and f, respectively.”
- ii. Line 223: “..(Fig.3f)

I would prefer you use "Goldschmidt tolerance factor" instead of "tolerance factor", at least when it is first introduced in the text.

Reviewer #3 (Remarks to the Author):

In this manuscript, the authors are trying to address a very important problem of antiferroelectrics, namely the observation of room temperature perfect polarization-field double hysteresis loops. For the following reasons, I do not think the authors achieved what they declared as promising results as documented in the manuscript. Thus I do not suggest its publication in Nature Communications.

- The authors mentioned “a well-defined double P-E loop with *near-zero* remnant polarization and significantly reduced hysteresis can be achieved when adding appropriate CH and AN (such as NN-20AN-5CH), demonstrating typical AFE characteristics.” Most of the hysteresis loops in the manuscript have a much larger plot range than the hysteresis height, which makes it extremely difficult to see the real value of the remnant polarization. For instance, Fig. 3d (the authors' best material NN-20AN-5CH) has a plot range between -60 and 60, while the hysteresis only ranges between -30 and 30. Such kind of plot is very

irresponsible and misleading.

- Reference [Tan X. et al., Double hysteresis loops at room temperature in NaNbO₃-based lead-free antiferroelectric ceramics, MATER. RES. LETT., 2018 VOL. 6, NO. 3, 159–164; <https://doi.org/10.1080/21663831.2017.1419994>] reported that the exposure of a modified NaNbO₃ ceramic to bipolar electric fields of ± 160 kV/cm at 100°C can preserve the double hysteresis loops at room temperature. These double hysteresis loops can still be observed after 125 days of room temperature aging with some decay in maximum polarization. The remnant polarization there can be as low as $2 \mu\text{C}/\text{cm}^2$. The authors should compare their results with some similar results. In addition, it would be better if the authors could comment on the lifetime of the so-called well-defined double hysteresis loops they observed.

Response Letter to Manuscript “Well-defined double hysteresis loop in NaNbO_3 antiferroelectrics” (NCOMMS-22-49159)

We would like to sincerely thank the reviewers for their time and effort in carefully reading the manuscript and preparing the review reports. We have revised our manuscript accordingly, the point-by-point responses to comments are enclosed. Our responses to the reviewers’ comments are highlighted in blue. In the revised manuscript, we highlighted our modifications in red. We hope we have satisfactorily addressed all reviewers’ concerns and questions.

Point-by-point responses to the reviewers’ comments

Reviewer #1 (Remarks to the Author):

The manuscript by Nengneg Luo et al. demonstrate a well-defined double hysteresis loop in NaNbO_3 antiferroelectrics ceramics by doping with CaHfO_3 and AgNbO_3 to stabilize the AFE P phase. The beautiful STEM images clearly confirmed the decreased cation displacements and BO_6 octahedral tilting angles, validating the theoretical prediction. The results are very interesting and important. I think that the manuscript is publishable in Nature Communications. Some small concerns are as follows:

Response:

We appreciate the reviewer’s very positive recommendation. Following the reviewer’s comments, we have revised the manuscript and the revisions are listed below.

Comment 1: The behaviors of the P-E hysteresis loops of the ceramic samples also depend on their grain size and morphology. The doping of AgNbO_3 and CaHfO_3 might change their microstructure obviously. In order to draw a solid conclusion, the authors should exclude the effect of grain morphology on its FE and AFE properties. So it is better to provide some SEM images or low mag TEM images.

Response:

Thanks for the valuable suggestion. According to the suggestion, we have added the SEM images of NN, NN-5CH and NN-20AN-5CH ceramics as shown in Fig. R1 (Supplementary Fig. 14). It can be seen all samples show equiaxed grains with closely compacted microstructure, except a few small particles (Hf-rich oxide) on the surface of matrix grains in CH modified NN-based ceramics. It should be noted the amount of

impurity can be neglected, because no detectable extra peak is observed in the XRD. The size distribution of most grains is around 5-10 μm , indicating the AN and CH modifications do not significantly change the grain size and morphology. All these results support the strong AFE features in NN-20AN-5CH ceramic are not associated with the changes in grain size and morphology.

In the revised manuscript, we have added the following sentences to explain there is no grain morphology effect on its FE and AFE properties: It should be noted that the strong antiferroelectricity can be ruled out from the effect of micromorphology, because both NN and CH modified NN-based ceramics exhibit similar grain size and morphology, though a small amount of Hafnium-rich oxide is observed in CH modified NN-based ceramics (Supplementary Figs. 14 and 15).

Fig. R1 SEM images of (a) NN, (b) NN-5CH, and (c) NN-20AN-5CH ceramics.
(Supplementary Fig. 14)

Comment 2: Do the doping elements (Ag, Ca and Hf) distribute uniformly? Is there any nanoscale segregation or phase separation?

Response:

Thanks for the valuable questions. We have added the SEM images and the corresponding elemental mapping images for the as-prepared NN-20AN-5CH ceramic, as given in Fig. R2 (Supplementary Fig. 15). Some small particles with white contrast were observed on the surface of the matrix grains. From the EDS, it can be found that Nb, Na, Mn, Ag and Ca are uniformly distributed in the samples. Majority of the Hf element is also uniformly distributed in the samples, but with very small amount of Hf-rich oxide aggregates in the form of small particles with size of 200-500 nm on the surface.

In the revised manuscript, we have added the following sentence to explain the doping distribution: It should be noted that the strong antiferroelectricity can be ruled out from the effect of micromorphology, because both NN and CH modified NN-based

ceramics exhibit similar grain size and morphology, though a small amount of Hafnium-rich oxide is observed in CH modified NN-based ceramics (Supplementary Figs. 14 and 15).

Fig. R2 SEM image and elemental mapping images of NN-20AN-5CH ceramic.
(Supplementary Fig. 15)

Comment 3: I want to know if the BF-STEM images shown in Fig. 1 d-f have been processed. The image contrast is a little strange for me.

Response:

Thanks for the careful reading and the valuable comment. The BF-STEM images in Fig. 1d-f in the manuscript are shown in the original dataset, only with a change in appearance color. A raw image of Fig. 1d in the typical grey scale is given in Fig. R3 for comparison. This imaging technique was developed by one of the co-author Dr Ye Zhu to map the octahedral tilt in complex perovskite oxides [Nature Materials 14, 1142–1149 (2015)]. With an optimized imaging condition, the octahedral tilt can be precisely mapped out. We agree that the contrast is strange because this technique primarily uses transmitted electrons to form the image, which heavily depends on the sample thickness, defocus and collection angle of the detector. The contrast may also undergo complex contrast reversal with respect to defocus and thickness [Ultramicroscopy 181 (2017) 1–7]. However, this technique can provide a wide window of defocus-thickness imaging parameters for localizing the light atoms (e.g. oxygen) in perovskite oxides, which is also less sensitive to sample mis-tilt [Ultramicroscopy 181 (2017) 1–7].

Fig. R3 Raw image of Fig. 1d in the typical grey scale.

Comment 4: In the caption of Fig.1, “The ADF images and structural schematics are overlaid in a, e, and f, respectively.” should be “are overlaid in d, e, and f, respectively.”

Response:

Thanks for the very careful reading and pointing out this careless mistake. The “The ADF images and structural schematics are overlaid in a, e, and f, respectively.” has been changed to “The ADF images and structural schematics are overlaid in d, e, and f, respectively.”, which is highlighted in the revised manuscript.

Reviewer #2 (Remarks to the Author):

Comment: The present work deals with the "design" (tailoring) of NaNbO_3 based ceramics with the aim of enabling field induced reversible phase transformations between the antiferroelectric and ferroelectric phases (which manifests itself in a corresponding well defined P-E hysteresis curve).

NaNbO_3 shows promising properties among lead-free antiferroelectric materials. However, although it has a seemingly simple chemical composition, it has long been regarded as probably the "most complex perovskite system known". This is due to the complex NbO_6 octahedral tilts that occur in the unit cell. These lead to a large number of different, partly (anti-)ferroelectric phases, with corresponding phase transitions between them. A special feature of NaNbO_3 is that at room temperature the energy difference between the ferroelectric Q-phase and the antiferroelectric P-phase is so small that both phases coexist and field-induced phase transformations can easily occur. In the case of NaNbO_3 , however, the application of an electric field leads to field-induced irreversible formation of the ferroelectric Q phase, resulting in a (broad) P-E hysteresis curve with a very high remanence. This property has so far prevented the technological use of pure NaNbO_3 as an antiferroelectric material.

This was the starting point of the present work. With the deliberate addition of AN (AgNbO_3) and CH (CaHfO_3), the NbO_6 tilts were reduced and thus the antiferroelectric P-phase could be stabilized. Moreover, the authors were able to demonstrate the reversible behavior of the such tailored ceramics in the P-E and I-E hysteresis curves. The (macroscopic) electrical investigations are thereby complemented by a very extensive structural characterization by means of X-ray diffraction and electron microscopy and supported by corresponding DFT calculations.

In my opinion, the authors impressively demonstrate with the present work the potential of the nowadays existing possibilities to tailor a material. This involves both pre-consideration of how the material should be designed to achieve the desired structural and physical properties, and the application of a variety of experimental techniques to verify the actual properties. Therefore, I believe that the present manuscript can be published in Nature Communications if the authors implement the following revisions.

Response:

We greatly appreciate the reviewer for acknowledging the importance and challenges of NN ceramic system and considering that we “impressively demonstrate with the present work the potential of the nowadays existing possibilities to tailor a material”, as well as very valuable and constructive comments on the manuscript. We will respond to the comments one by one below.

Minor comments

Comment 1: In the literature, the coexistence of long-range antiferroelectric and ferroelectric phases is – on a microscopic scale - discussed to be caused by irregularly arranged nanotwins resulting from complex local octahedral tilt patterns (e.g., Johnston 2010, <https://doi.org/10.1021/ja101860r>; Jiang 2013, <https://doi.org/10.1103/PhysRevB.88.014105>). Can you please comment on this for the present material system?

Response:

Thanks for the good suggestion and providing the relevant references, which are very useful to our understanding of the NN microstructure.

The reference (Jiang 2013, <https://doi.org/10.1103/PhysRevB.88.014105>) used the pair distribution function (PDF) and high-resolution powder neutron diffraction to study the local structure and average crystal structure. They stated that “the long-range structure with *Pbcm* space group is stacked by small *R3c* domains within the short range below 5.2 Å, by rotating the a and b axes by 45° from those of the *R3c* unit cell”. This means the *Pbcm* symmetry can be built along the diagonal direction of *R3c* structure by twinning operation. Furthermore, this requires the local domains in the *R3c* group have an anti-parallelled spontaneous polarization to revolve into the *Pbcm* group. The length of the *Pbcm* unit cell ($c = 15.48 \text{ \AA}$) is about twice as the *R3c* unit cell ($c = 7.82 \text{ \AA}$) and about four times as the elementary cubic cell (3.95 \AA), so every basic polarized cell also forms similar twin structures to make the polarizations stack antiparallely through the whole structure.

The reference (Johnston 2010, <https://doi.org/10.1021/ja101860r>) mentioned that “the mechanism of octahedral tilting within the AFE *Pbcm* structure is effectively a

“twinning” operation when compared with FE $P2_1ma$ structure”. Hence the standard a^-b^+ system in $P2_1ma$ becomes $[a^-b^+/a^-b^-/a^-b^+]$ via an inversion about the central block (equivalent to an additional out-of-phase tilt between blocks 2 and 3 along the c -axis). The AFE $Pbcm$ structure also possesses a longer c axis about four times as the elementary cubic cell.

In the P phase with $Pbcm$ space group, the octahedral rotation sequence along b axis is repeated every four octahedra, as expressed by AACCAACC..., where A and C represent anticlockwise and clockwise respectively. The local AA or CC sequence means the existence of local a^-b^+ tilting with the FE Q phase structure. A twinning operation thus takes the form of AC sequence. In the same manner, if we focus onto the local AC or CA sequence, which points to the local a^-a^- tilting of the $R3c$ space group, the AFE phase can then be regarded as a twinning structure of local $R3c$ domains. Even though the above two references use different space group for FE phase, they derive a similar concept that the long-range AFE structure can be stacked by a local scale “twinning” operation of FE phase.

The newly designed NN-based AFE materials exhibit AFE structure with $Pbcm$ space group, but cannot be simply regarded as a twinning operation of the FE structure. First, the local structure is highly ordered with sharp $n/4$ reflection spots in the XRD and SAED patterns, in contrast to pure NaNbO_3 . Such modulation is a characteristic feature for antiferroelectric phase. Second, the antiferroelectric structure is thermodynamically stable and can be fully recovered after removal of the electric field, which is also significantly different from the FE twins. This suggests other factors, especially the correlation between the anti-paralleled polarization in the adjacent crystal lattice, may also be responsible for the recoverable AFE phase in the newly designed NN-based AFE materials. We believe that the structure of AFE is very complex, especially for NN system, we hope more comprehensive studies will be conducted to understand the structure origin.

The above two references show experimental results of coexistence of two phases in pure NN, even though they use difference space groups ($Pbcm$ and $P2_1ma$; $Pbcm$ and

R3c). In order to show more experimental results of coexistence of two phases in pure NN, the other reference (Johnston 2010, <https://doi.org/10.1021/ja101860r>) has been also added in the introduction part in the revised manuscript.

Comment 2: The manuscript is written very compactly, i.e. the reader has to "process" very much information in a few sentences. Due to this high data density (especially also in the supplementary), one has to search very often in the text for the corresponding figures. In my opinion, this makes the flow of the text somewhat difficult. The high density of information also means that the figures often appear overloaded and individual diagrams are sometimes also too small to still be able to recognize important details. I would therefore recommend to revise the illustrations in this respect and also to think about a corresponding revision of the text. In addition, the manuscript contains occasional language errors, but these can be easily corrected.

Response:

Thanks for the valuable and constructive suggestion. We totally understand the high data density of our paper and the difficulty of reading the paper. To make the manuscript more readable, based on the reviewer's suggestion, we have made the following changes in the revised manuscript. (1) Much of the description and discussion of the figures in the Supplementary Information has been simplified in the manuscript and moved to the figure captions, we only kept the main conclusions based on these figures in the main text. These include the XRD, Raman spectra, and temperature dependence of dielectric spectra. (2) The sequence of Figs.3a-h and the Supplementary Figs. 11-13 have been adjusted to improve context continuity for easy reading. (3) The scale ranges for polarization and current in P - E loops and I - E curves in Figs.3a-h have been modified to make it easier for the readers to assess the values. (4) The text size has been adjusted and some new markers have also been added in some figures to make them more readable. Figs. R4 and R5 are two examples of the modifications for Figs. 1 and 5 respectively in the revised manuscript. (5) We have thoroughly read our manuscript to avoid the careless language errors.

We hope the above changes will allow for easier reading and contextual continuity in the manuscript.

Fig. R4 Structure and DFT calculations of NN.

(Fig. 1)

Fig. R5 Polarization field mapping.

(Fig. 5)

Comment 3: Besides occasional linguistic errors I found the following minor errors:

i. Line 145: “.. are overlaid in d, e, and f, respectively.”

ii. Line 223: “..(Fig.3f)

Response:

Thanks for the very careful reading and pointing out the careless mistakes. We have corrected the linguistic errors and also thoroughly read the manuscript to avoid similar mistakes.

Comment 4: I would prefer you use "Goldschmidt tolerance factor" instead of "tolerance factor", at least when it is first introduced in the text.

Response:

Thanks for the good suggestion. The phrase “tolerance factor” is replaced by “Goldschmidt tolerance factor” when it is first appeared in Abstract and Introduction part.

Reviewer #3 (Remarks to the Author):

In this manuscript, the authors are trying to address a very important problem of antiferroelectrics, namely the observation of room temperature perfect polarization-field double hysteresis loops. For the following reasons, I do not think the authors achieved what they declared as promising results as documented in the manuscript. Thus I do not suggest its publication in Nature Communications.

Comment 1: The authors mentioned ``a well-defined double P-E loop with *near-zero* remnant polarization and significantly reduced hysteresis can be achieved when adding appropriate CH and AN (such as NN-20AN-5CH), demonstrating typical AFE characteristics." Most of the hysteresis loops in the manuscript have a much larger plot range than the hysteresis height, which makes it extremely difficult to see the real value of the remnant polarization. For instance, Fig. 3d (the authors' best material NN-20AN-5CH) has a plot range between -60 and 60, while the hysteresis only ranges between -30 and 30. Such kind of plot is very irresponsible and misleading.

Response:

Thanks for the good comments. We used a larger scale of -60 to 60 for consistency across all the figures- Fig. 3a to Fig. 3d, for a direct comparison of the four representative compositions and a good eye guide. The exact value of remnant polarization (P_r) is 1.7 $\mu\text{C}/\text{cm}^2$ for NN-20AN-5CH, which has been discussed in the text, this is actually a very low value among all lead-free AFE material, especially for NN-based system. For example, AgNbO_3 -based AFEs has P_r value of 1.7-5 $\mu\text{C}/\text{cm}^2$ (J. Mater. Chem. A, 2016, 4, 17279; Adv. Mater. 2017, 29, 1701824 & Adv. Mater. 2017, 29, 1701824). As for NaNbO_3 based AFEs, the P_r was reported to be in the range of 5-20 $\mu\text{C}/\text{cm}^2$ (Acta Materialia, 2020, 200: 127–135, Chem. Mater. 2021, 33, 266–274, Acta Materialia, 2021, 208: 116710, J. Appl. Phys. 2016, 120, 204102; Mater. Res. Lett., 2018, 6 (3): 159–164). Of particular importance is that a larger FE-AFE phase transition electric field exceeding 100 kV/cm is achieved in NN-20AN-5CH with a significantly reduced hysteresis, which has never been achieved over the past 70 years.

Following the reviewer's comment, we have changed the scale to -50 to 50 to have a better view of the remnant polarization. In addition, the specific values of remnant polarization and other electrical properties of NN-20AN-5CH are compared to other compositions and listed in Supplementary Table 2 in the Supplementary Information.

Comment 2: Reference [Tan X. et al., Double hysteresis loops at room temperature in NaNbO₃-based lead-free antiferroelectric ceramics, MATER. RES. LETT., 2018 VOL. 6, NO. 3, 159–164; <https://doi.org/10.1080/21663831.2017.1419994>] reported that the exposure of a modified NaNbO₃ ceramic to bipolar electric fields of ± 160 kV/cm at 100°C can preserve the double hysteresis loops at room temperature. These double hysteresis loops can still be observed after 125 days of room temperature aging with some decay in maximum polarization. The remnant polarization there can be as low as 2 $\mu\text{C}/\text{cm}^2$. The authors should compare their results with some similar results. In addition, it would be better if the authors could comment on the lifetime of the so-called well-defined double hysteresis loops they observed.

Response:

Thanks for the valuable suggestion and providing the relevant reference for us to study. The results reported in the reference showed the fresh (Na_{0.95}Li_{0.01}Ca_{0.04})(Nb_{0.96}Zr_{0.04})O₃ ceramic didn't acquire double *P-E* loop at room temperature, but were observed after high-temperature electric field treatment, as shown in Fig. R6 (a) and (b). It is worth noting that the double *P-E* loop obtained at room temperature after specific treatment also demonstrated a high remnant polarization of 9 $\mu\text{C}/\text{cm}^2$, while the low remnant polarization of 2 $\mu\text{C}/\text{cm}^2$ can only be observed at specific condition (long time aging) but with a very low maximum polarization of 9.7 $\mu\text{C}/\text{cm}^2$.

In comparison, the newly designed NN-20AN-5CH ceramic shows well-defined *P-E* loop with low remnant polarization of 1.7 $\mu\text{C}/\text{cm}^2$, high AFE-FE and FE-AFE phase transition electric field (>100 kV/cm), and greatly reduced hysteresis at room temperature. The well-defined double *P-E* loops have been observed over temperature range of 25-150 °C, all of which exhibit very low remnant polarization and very high phase transition electric fields, with greatly reduced hysteresis at elevated temperatures, as shown in Fig. R6c.

It is worth mentioning that the high-temperature electric treatment method proposed

in the reference can introduce double P - E loop in NN-based material, can be regarded as one approach to realize double P - E loop. We have added this paper as a reference and discussed this in the introduction of the revised manuscript.

As for the lifetime of the double hysteresis loops, the reported $(\text{Na}_{0.95}\text{Li}_{0.01}\text{Ca}_{0.04})(\text{Nb}_{0.96}\text{Zr}_{0.04})\text{O}_3$ ceramic in the given reference demonstrated “double hysteresis loops can still be observed after 125 days of room temperature aging with some decay in maximum polarization”, but this material still exhibits large remnant polarization, low FE-AFE phase transition electric field, and large hysteresis. Regarding the lifetime of our studied AFE ceramics, on the contrary, the double P - E loops of NN-20AN-5CH ceramic can be well preserved after more than 180 days with nearly no change, as shown in Fig. R6d, revealing a very stable performance with aging time.

According to this comment, we have added the reference in our manuscript and the relevant discussions, we have also added Fig. R6d, the aging behavior of the P - E loops for NN-20AN-5CH ceramic, as Supplemental Fig. 11. We use “... as well as good aging related performance over 180 days (Supplementary Fig. 11).” in the manuscript to describe the good aging time stability.

Fig. R6 (a) Hysteresis loops recorded under ± 160 kV/cm during the heating sequence, (b) Hysteresis loops recorded during the cooling sequence. The Figures are acquired from reference (Tan X. et al., MATER. RES. LETT., 2018, 6(3):159–164). (c) Temperature and (d) aging time dependence of P - E loops for NN-20AN-5CH ceramic.

REVIEWERS' COMMENTS

Reviewer #1 (Remarks to the Author):

The revised manuscript is ok for me.

Reviewer #2 (Remarks to the Author):

In my opinion, the authors made the changes I requested in the manuscript and also answered my questions satisfactorily. As far as I can judge, this also applies to the corresponding answers to the reports of the other two reviewers.

For this reason, I strongly support publication of the present manuscript in Nature Communications.

Reviewer #3 (Remarks to the Author):

Thanks for the clarification from the authors. My previous concerns about the remnant polarization are resolved. I can now understand the authors were trying to keep consistency across the subplots. For that, I guess a zoom-in inset could be helpful to show near-zero remnant polarization better.

As I have mentioned in my previous comments, what the authors are trying to address is a very important topic in this material; and now I believe they indeed has achieved novel results. I would like to recommend this manuscript be published in Nature Communications.

Response Letter to Manuscript “Well-defined double hysteresis loop in NaNbO₃ antiferroelectrics” (NCOMMS-22-49159A)

We would like to sincerely thank the reviewers for their time and effort in carefully reading the manuscript and preparing the review reports. We have revised our manuscript accordingly, the point-by-point responses to comments are enclosed. Our responses to the reviewers' comments are highlighted in blue. In the revised manuscript, we highlighted our modifications in red. We hope we have satisfactorily addressed all reviewers' concerns and questions.

Point-by-point responses to the reviewers' comments

Reviewer #1 (Remarks to the Author):

Comment:

The revised manuscript is ok for me.

Response:

Thanks to the reviewer's positive comment.

Reviewer #2 (Remarks to the Author):

Comment:

In my opinion, the authors made the changes I requested in the manuscript and also answered my questions satisfactorily. As far as I can judge, this also applies to the corresponding answers to the reports of the other two reviewers.

For this reason, I strongly support publication of the present manuscript in Nature Communications.

Response:

Thanks to the reviewer's positive comment. We really appreciate the recommendation to publish the manuscript in Nature Communications.

Reviewer #3 (Remarks to the Author):

Comment:

Thanks for the clarification from the authors. My previous concerns about the remnant polarization are resolved. I can now understand the authors were trying to keep consistency across the subplots. For that, I guess a zoom-in inset could be helpful to show near-zero remnant polarization better.

As I have mentioned in my previous comments, what the authors are trying to address is a very important topic in this material; and now I believe they indeed has achieved novel results. I would like to recommend this manuscript be published in Nature Communications.

Response:

Thanks to the reviewer's positive comment. In order to make the readers easier to evaluate the near-zero remnant polarization of NN-20AN-5CH ceramic, we have added a magnified scale for this composition as an inset in Figure 4a in the revised manuscript,

as shown in Figure R1.

Figure R1 the P - E loops of the as designed NN-based ceramics. The inset shows magnified scale for NN-20AN-5CH.

(Figure 4a in the revised manuscript)